Identification of key genes in sepsis-induced cardiomyopathy based on integrated bioinformatical analysis and experiments in vitro and in vivo

Liu Dehua 1
Wang Tao 2
Wang Qingguo 2
Dong Peikang 2
Liu Xiaohong 2
Li Qiang 2
Shi Youkui 3
Li Jingtian 2
Zhou Jin 4 zhoujin@wfmc.edu.cn
Zhang Quan 2 fyzhangquan@wfmc.edu.cn
1 Weifang Medical University , Weifang , China
2 Department of Cardiology, Affiliated Hospital of Weifang Medical University , Weifang , China
3 Department of Emergency Medicine, Affiliated Hospital of Weifang Medical University , Weifang , China
4 School of Pharmacy, Weifang Medical University , Weifang , China
Li Tian
Electronic publication date: 2023 Nov 21
Publication date: 2023
Volume: 11
Electronic Location ID: e16222
Received 2023 May 11; Accepted 2023 Sep 11
Copyright: © 2023 Liu et al.
Copyright year: 2023
Copyright holder: Liu et al.
License: This is an open access article distributed under the terms of the Creative Commons Attribution License, which permits unrestricted use, distribution, reproduction and adaptation in any medium and for any purpose provided that it is properly attributed. For attribution, the original author(s), title, publication source (PeerJ) and either DOI or URL of the article must be cited.
License URL: https://creativecommons.org/licenses/by/4.0/

Keywords: Sepsis-induced cardiomyopathy, TPT1, PPI, Hub gene

Funding: National Natural Science Foundation of China 82100376 This work was supported by the National Natural Science Foundation of China (No. 82100376). The funders had no role in study design, data collection and analysis, decision to publish, or preparation of the manuscript.

==============================
Introduction

Sepsis is a life-threatening disease that damages multiple organs and induced by the host’s dysregulated response to infection with high morbidity and mortality. Heart remains one of the most vulnerable targets of sepsis-induced organ damage, and sepsis-induced cardiomyopathy (SIC) is an important factor that exacerbates the death of patients. However, the underlying genetic mechanism of SIC disease needs further research.

Methods

The transcriptomic dataset, GSE171564, was downloaded from NCBI for further analysis. Gene expression matrices for the sample group were obtained by quartile standardization and log2 logarithm conversion prior to analysis. The time series, protein-protein interaction (PPI) network, and functional enrichment analysis via Gene Ontology and KEGG Pathway Databases were used to identify key gene clusters and their potential interactions. Predicted miRNA-mRNA relationships from multiple databases facilitated the construction of a TF-miRNA-mRNA regulatory network. In vivo experiments, along with qPCR and western blot assays, provided experimental validation.

Results

The transcriptome data analysis between SIC and healthy samples revealed 221 down-regulated, and 342 up-regulated expressed genes across two distinct clusters. Among these, Tpt1, Mmp9 and Fth1 were of particular significance. Functional analysis revealed their role in several biological processes and pathways, subsequently, in vivo experiments confirmed their overexpression in SIC samples. Notably, we found TPT1 play a pivotal role in the progression of SIC, and silencing TPT1 showed a protective effect against LPS-induced SIC.

Conclusion

In our study, we demonstrated that Tpt1, Mmp9 and Fth1 have great potential to be biomarker of SIC. These findings will facilitated to understand the occurrence and development mechanism of SIC.

Introduction

Sepsis is a systemic inflammatory response caused by bacteria, fungi, viruses, or parasites and is life-threatening (Lu et al., 2018). Heart injury is pivotal symptom in sepsis (Suzuki et al., 2017), as it significantly hampered overall blood circulation, along with tissue hypoxia and mitochondrial and metabolic dysfunction. Sepsis-induced cardiomyopathy (SIC) first described by Parrillo et al. (1985) is an acute cardiac dysfunction induced by sepsis (Krishnagopalan et al., 2002) and largely contributes to increased morbidity and mortality rates among sepsis patients. Previous reports revealed that prevalence of SIC ranging from 13.8 to 51.6% and mortality rate reaching approximately 70% (Drosatos et al., 2015; Flynn, Mani & Mather, 2010). SIC was originally described as an acute decrease in left ventricular ejection fraction with ventricular dilation in patients with sepsis (Wang et al., 2020). At present, most researchers define it as an acute cardiac dysfunction syndrome in sepsis patients not associated with myocardial ischemia (Hennein et al., 1994). SIC has a variety of manifestations, including systolic or diastolic left and/or right ventricle damage, cardiac output, and oxygen deficiency, or primary myocardial cell damage (Yalta, Yilmaztepe & Zorkun, 2018). However, it lacks unified definition to better describe them, additionally, the pathogenesis of SIC has not been fully elucidated and the diagnostic criteria have not been unified, leading slow progress in relevant research and treatment (Tfelt-Hansen & Koehler, 2011).

Several biomarkers have been associated with SIC. Troponin I (cTnI), a specific and sensitive marker of myocardial injury (Adams et al., 1993), shows increased levels in SIC patients and is directly related to myocardial cell injury severity and disease progression (Liu et al., 2021). A small sample study revealed a positive correlation between heart rate and cTnI in children with septic shock. Due to the association of myocardial damage to myocardial infarction, kidney damage, poisoning, and other serious patients, as a sensitive marker of myocardial injury cTnI in early identification of SIC and prognostic evaluation lack of specificity, cannot be used as an ideal SIC laboratory index (Oras et al., 2015). Type B natriuretic peptide (BNP), a sensitive marker of cardiac dysfunction, reflects the left ventricular systolic function, but a certain extent reflects the left ventricular diastolic function and right ventricular function (Gatzoulis et al., 1995). Research shows that BNP in critically sick individuals is unrelated to left ventricular filling pressure, pulmonary artery wedge pressure, or cardiac index (Greenberg et al., 1979). Sepsis patients may indicate a large rise in BNP but no evident cardiac function impairment (Maeder et al., 2006). A survey of 900 patients with septic shock (with or without SIC) found that BNP in critical patients is not correlated with pulmonary wedge pressure, left ventricular filling pressure, and cardiac index (Scerbo & Moore, 2017). NT-proBNP and CINT are closely related to the occurrence of septic shock (Thorburn, 2010). Therefore, it is urgent to identified more specific SIC biomarkers.

The majority of research indicates that SIC is a structurally normal and reversible process of cardiac injury (Umezawa et al., 2020). At present, clinical diagnosis is mainly assisted by cardiac ultrasound, but there still exist some gaps, and it is difficult for common laboratory indicators to exhibit high specificity and sensitivity (Abboud et al., 1996). SIC treatment lacks evidence-based recommendations, and treatment for the primary disease, sepsis remains the only option (Sartelli et al., 2013). Human trials of treatments that target inflammatory cytokines have not proven effective (Meng et al., 2020). Therefore, only by fully understanding the pathophysiological mechanism of SIC and adopting scientific and effective prevention and treatment measures are expected to reduce the incidence and mortality of SIC. This study aims to identify the genetic mechanism and candidate biomarkers of SIC. We try to cluster genes into different clusters to screen for genes that are consistently up-regulated and down-regulated. Functional and PPI analyses were also performed to narrow down the key genes. Our results may highlight new causal candidate gene sites for treating SIC.

Materials and methods

Data sources

Gene Expression Omnibus (GEO, http://www.ncbi.nlm.nih.gov/geo/) database (Clough & Barrett, 2016) was used to download the expression matrix associated with SIC. GSE171546 dataset contained 20 mice treated with cecal ligation and puncture at 0, 24, 48, and 72 h. RNA was extracted from the heart tissues, and mRNA expression was detected by Illumina NovaSeq 6000.

Data pre-processing

Firstly, the probe ID was replaced with gene symbols using the platform annotation information table. Genes with same symbol were incorporated. Expression matrixes samples were obtained according to ID. We use normalize between arrays in the limma (Smyth, 2005) package to standardize the quartile of the obtained chip expression data. Then, the ID of multiple expressions of the same gene was calculated as the average expression amount of the gene.

Timing analysis

Time series transcriptome data reflect the expression of genes at different moments. It can more accurately depict the gene expression levels of different biological processes and stages of the same biological process, which is more in line with the actual situation and is of great significance for studying the dynamics and diversity of biological processes. Mfuzz (Kumar & Futschik, 2007) clusters time series data based on a fuzzy clustering algorithm. Continuous up and down-regulation genes were selected for subsequent analysis according to the results of time series analysis.

Protein-protein interaction (PPI) network forecast

Hub genes in selected clusters were analyzed using String (Mering et al., 2003) online tool for PPI analysis. A combined score greater than 0.9 is chosen as the threshold for protein-protein interaction. Based on the adequate protein-protein interaction (PPI) relationship pairs, Cytoscape (Shannon et al., 2003) was then selected to perform the topology of the PPI relationship network. Most biological networks obey the attribute of a scale-free network. Therefore, the important node involved in the protein interaction relationship in the PPI network, namely hub protein, can be obtained using the connectivity degree analysis in network statistics. In this article, the nodes of the interaction networks were analyzed, and the scale-free nature of the interaction protein networks found the central proteins in the network.

GO and KEGG analysis

Functional enrichment analysis based on candidate genes was conducted using the Gene Ontology Database (Gene Ontology Consortium, 2004) and KEGG Pathway Database (Kanehisa, 2002). Fisher’s exact test is used to find out which specific functional items have the greatest correlation with a set of genes. Each item corresponds to a statistical p-value to indicate significance in the analysis results. The smaller the p-value, the higher probability of having a correlation between the item and the input gene. That is, most of the genes in this group have the description function corresponding to this item.

Prediction of mRNA target miRNA

Five miRNA databases, such as miRanda (Turner, 1985), miRDB (Chen & Wang, 2020), TargetScan (Edris, 2011), and miTarBase (Chou et al., 2018), were used to predict the relationships between mRNAs and miRNAs. The miRNA-mRNA relationships retrieved from at least two databases were selected for the subsequent construction of the TF-miRNA-mRNA regulatory network. According to literature-level TF-miRNA regulation data based on different species in the TransmiR v2.0 database (http://www.cuilab.cn/transmir), relationships between miRNA and TF were found. According to the relationship among miRNA-mRNA, miRNA-TF, and mRNA-TF, cytoscape was used to construct a TF-miRNA-mRNA regulatory network.

Animal model

C57BL/6J mice weighing 18–25 g were purchased from SLAC Laboratory Animal Co., Ltd. (Shanghai, China). These mice were housed in individually ventilated cages with ad libitum access to standard laboratory chow and water. The Ethical Committee of the Affiliated Hospital of Weifang Medical University granted ethical approvals for animal experiments (#2022WFM017). The Ethical Committee ratified all surgical procedures. Briefly, these mice were firstly anesthetized by intraperitoneal injection of pentobarbital (50 mg/kg), then they were fixed in the supine position, and the skin of the operation area was routinely disinfected. Finally, these mice were randomly divided into the Sham group and the treatment group (n = 6). As control, we performed cecal isolation and abdominal closure surgery on mice of the Sham group. In the treatment group, the distal cecum of mice was opened, lapped at 0.5 cm, and closed. After operation, all mice were reared in separate cages and fed freely. CO2 inhalation was employed at the end of the experiment, following established guidelines.

Real-time quantitative PCR

TRIzol reagent (T9424; Sigma-Aldrich, Beijing, China) was used to extract RNA from heart tissues according to the manufacturer’s protocol. Then 1 μg RNA was used to synthesize cDNA, followed by gene expression analysis on ABI 7300 qPCR system. Relative mRNA levels were determined after normalization using GAPDH or U6 as an internal control.

Western blot

Proteins were extracted from heart tissue with RIPA lysis buffer (Beyotime Biotechnology). Protein samples (60 μg) were separated by SDS-PAGE electrophoresis and transferred to PVDF membranes (Millipore). After blocking, protein on the membrane was incubated with primary antibodies such as MMP-9 (NBP2-13173; Novus Biologicals, Centennial, CO, USA), TPT1 (PA5-34503) and FTH1 (NBP1-31944, Novus Biologicals) at 4 °C overnight. Next day they wereincubated with HRP-conjugated secondary antibody. FluorChemE imager (Alpha) was used for visualization, and the expression level of specific protein was normalized to GAPDH level.

Verifying the involvement of TPT1 in the development of SIC

Hl-1 myocardial cell line of mice purchased from the American Type Culture Collection was used for further verification. In brief, cells were cultured in DMEM medium containing 15% newborn bovine serum at the density of 3 × 106 cells per bottle. Cells were firstly inoculated in culture bottles with a base area of 75 cm2, then the experiment was carried out after 24 h, in this step cells should culture in DMEM medium without newborn bovine serum. Lipopolysacchride (LPS, 25 mg·L−1) was used to induced myocardial cell injury model for 24 h. The protein expression levels of TPT1 and GAPDH were detected by western blot at 6, 12 and 24 h after LPS treatment.

Another experiment was performed to evaluate the role of TPT1 in development of SIC. The small interfering RNA for TPT1, 5′-AAGGTACCGAAAGCACAGTAA-3′ (siRNA1), or 5′-AACCATCACCTGCAGGAAACA-3′ (siRNA2) were synthesized by Jima Pharmaceutical Technology Co., Ltd. China and siRNA duplex 5′-AACCATCACTTACAAGAAACC-3′ was used as control. Hl-1 cell line was cultured in DMEM medium for 24 h and treated differently for 24 h; LPS group: Hl-1 myocardial cell line was treated with LPS (25 mg·L−1); si-TPT1 group: Hl-1 cell line with TPT1 silencing was treated with LPS (25 mg·L−1); LPS+NC group: Hl-1 cell line was cultured in DMEM medium with LPS (25 mg·L−1). Then, the apoptotic cells were detected using TdT-mediated dUTP Nick-End Labeling (TUNEL). Flow cytometry was used to detect the cell cycle characteristics from the four groups.

Statistical analysis

Statistical analysis was performed using a Mann-Whitney tests, unpaired two-tailed Student’s t-test or one-way ANOVA. Statistical significance is denoted as p value smaller than 0.05. In all graphs standard error of the mean (SEM) is calculated and error bars are plotted according to mean ± SEM.

Results

Continuous up-regulation and down-regulation gene screening

Mfuzz was used to cluster time-series data based on a fuzzy clustering algorithm. K-nearest neighbor weight algorithm (KNNW) was selected for the algorithm. The standard deviation of gene screening was 0.25, and the neighborhood coefficient membership was 0.5. Finally, six gene clusters were obtained, and genes in cluster1 and cluster3 were selected for further analysis (Fig. 1). In cluster1, 221 genes showed a continuous down-regulation expression trend. Among them, the first five genes in coefficient membership were Pm20d2, Ybx2, Hacd1, Clasp1, and Ecrg4 (Table 1). As for cluster3, 342 up-regulation genes were obtained, which contained Mmp8, Rps13, Rps15a, Plp2, and Vps51.

Figure 1 (A–F) Timing analysis of the genes in GSE171546 datasets.

A total of six gene clusters were obtained. Cluster1 and cluster3 were further analyzed.

Table 1 The top 20 genes in cluster1 and cluster3 based on the coefficient membership.

Gene in Cluster1	MEM.SHIP	Gene in cluster3	MEM.SHIP	
Pm20d2	0.853839853	Mmp8	0.866586146	
Ybx2	0.849720181	Rps13-ps1	0.863721432	
Hacd1	0.847773469	Rps15a	0.862700716	
Clasp1	0.846836573	Plp2	0.861445711	
Ecrg4	0.846260406	Vps51	0.860814299	
Itgb6	0.841702114	Ubxn6	0.857765509	
Lactb	0.831414882	Washc1	0.854778181	
Lynx1	0.83004354	mt-Co1	0.854347334	
Scgb1c1	0.827212546	Rpl14	0.852620451	
Afg1l	0.825915331	Rps13	0.849763108	
Actc1	0.825720804	Spp1	0.846487716	
2310020H05Rik	0.823928702	Spg11	0.84511117	
Fblim1	0.821182942	Rpl19	0.844841583	
Lingo3	0.819656264	Rps9	0.842416394	
Gm10635	0.817260004	Tmem43	0.837706237	
Ptcd3	0.806533597	Rps8	0.835137519	
Cfd	0.806463425	Faf1	0.833398647	
Gm12319	0.804066657	Bcl2l1	0.832042017	
Pxdn	0.801408788	Gstm1	0.831836914	
C130080G10Rik	0.801194925	Eif4b	0.829987869	

Functional analysis of the genes in cluster1 and cluster3

A total of 563 genes of the continuous up-regulated and down-regulated clusters were selected to perform the GO and KEGG analysis. Figure 2A and Table 2 revealed that the genes mainly enriched in several biological processes (BP) terms such as Translation, Peptide biosynthetic process, Peptide metabolic process, Amide biosynthetic process and Gene expression. As for cellular component (CC) terms, the cytosolic ribosome, ribosome, cytosolic large ribosomal subunit, large ribosomal subunit and cytosolic small ribosomal subunit were shown in Fig. 2B. Figure 3C showed the significant enriched molecular function (MF) terms, such as rRNA binding, RNA binding, 5S_rRNA binding, ubiquitin-protein transferase regulator activity and translation factor activity RNA binding.

Figure 2 Functional analysis based on the GEO datasets in this study.

GO analysis: (A) biological process; (B) cellular component; (C) molecular function. (D) KEGG analysis.

Table 2 Statistics of the gene counts and p-value of BP, CC, and MF terms.

Gene in Cluster1	MEM.SHIP	Gene in cluster3	MEM.SHIP	
Pm20d2	0.853839853	Mmp8	0.866586146	
Ybx2	0.849720181	Rps13-ps1	0.863721432	
Hacd1	0.847773469	Rps15a	0.862700716	
Clasp1	0.846836573	Plp2	0.861445711	
Ecrg4	0.846260406	Vps51	0.860814299	
Itgb6	0.841702114	Ubxn6	0.857765509	
Lactb	0.831414882	Washc1	0.854778181	
Lynx1	0.83004354	mt-Co1	0.854347334	
Scgb1c1	0.827212546	Rpl14	0.852620451	
Afg1l	0.825915331	Rps13	0.849763108	
Actc1	0.825720804	Spp1	0.846487716	
2310020H05Rik	0.823928702	Spg11	0.84511117	
Fblim1	0.821182942	Rpl19	0.844841583	
Lingo3	0.819656264	Rps9	0.842416394	
Gm10635	0.817260004	Tmem43	0.837706237	
Ptcd3	0.806533597	Rps8	0.835137519	
Cfd	0.806463425	Faf1	0.833398647	
Gm12319	0.804066657	Bcl2l1	0.832042017	
Pxdn	0.801408788	Gstm1	0.831836914	
C130080G10Rik	0.801194925	Eif4b	0.829987869	

Figure 3 PPI network.

PPI network based on the genes that are continuously up-regulated and down-regulated.

KEGG analysis was also analyzed and the results were shown in Fig. 2D. Ribosome, lysosome, glycolysis/gluconeogenesis, valine leucine and isoleucine degradation, and carbon metabolism pathways were focused on.

PPI network analysis

PPI network analysis utilizing the string database and a combined score threshold of 0.9 was used to identify SIC hub genes. As Fig. 3 shown, Rps9, Rps15a, Rps27, Rps3, Rps5 and Rps6 had the highest connectivity with 66 and 65 other proteins, respectively. In addition, complex interactions among Rps13, Rps14, Rps23 and Rps25 were also found, indicating potential functional groups for SIC. The list of the top 20 genes of the PPI network is shown in Table 3.

Table 3 Statistics of the top 20 genes of the PPI network based on the degrees.

Gene	Degree	Gene	Degree	Gene	Degree	Gene	Degree	
Rps9	66	Rps6	65	Rps19	63	Fau	62	
Rps15a	65	Rps13	64	Rps28	63	Rps18	62	
Rps27	65	Rps14	64	Rps3a1	63	Rpl4	61	
Rps3	65	Rps23	64	Rps7	63	Rpl8	61	
Rps5	65	Rps25	64	Rps8	63	Rps20	61	

Statistics of the target miRNAs of the selected genes

The continuously up-regulated and down-regulated genes were selected to predict the target miRNAs using five miRNA databases. As Fig. 4A shown, 155 miRNA-mRNA pairs were found in miTareBase and miRanda databases, and 1,340 miRNA-mRNA pairs were in miRDB and TargetScan. A total of 2,900 miRNA-mRNA pairs verified at least two datasets were selected for further analysis. Next, the miRNAs were used to predict the TF-miRNA regulated pairs using TransmiR v2.0 database and 2,763 TF-miRNA pairs were obtained. Furthermore, the Trrust database was used to predict the TF-mRNA pairs, generating a total of 109 TF-mRNA pairs.

Figure 4 Conduction of the TF-miRNA-mRNA network using genes in the two selected clusters.

(A) Venn plot of the miRNA-Mrna pairs in the four databases. (B) The TF-miRNA-mRNA network. Rhomboid is miRNA, the orange circle is up-regulated gene, and the inverted triangle is TF.

Regulatory network of TF-miRNA-mRNA

By predicting the regulatory relationship of mRNA and miRNA, a network with 143 regulatory relation pairs was conducted (Fig. 5). Among the down-regulated genes, Ptp4a1, Pth and Nt5e owned the most pairs of regulatory relationships, which were 26, 10 and 10, respectively. As for up-regulated genes, Fam126b, Ebf1, St3gal6 and Hmgn3 interacted with 16, nine, seven and seven miRNAs, respectively. According to the statistical data, miR-294 regulated three genes simultaneously, such as Ddhd1, Cxadr, and Ptp4a1. According to the relationship among miRNA-mRNA, TF-miRNA and TF-mRNA above, the TF-miRNA-mRNA regulatory network, which contained three subnetworks, was conducted, as Fig. 4B shown. The left subnetworks indicated that the TF Rela directly regulated the expression of Fth1 and indirectly regulated Fth1 by miR-150-3p. In the middle subnetwork, TF Smad3 promoted the expression of Mmp9 and Tpt1 or accomplished it indirectly by regulating miR-323-5p and miR-654-5p, respectively. Gene Bcl2l1 was related to miR-342-3p. miR-342-5p and TF Stat6.

Figure 5 (A and B) PPI networks based on the genes in TF-mRNA pairs.

The orange circles represent up-regulated genes. The blue circles represent down-regulated genes. The octagon represents TFs.

In addition, the TF-mRNA regulatory pairs were also selected for PPI network analysis. As shown in Fig. 5A, the TF Tpt1 promoted the expression of 69 genes, mainly belonging to the Rps and Rpl gene families. The up-regulated expression of Mmp8, Ctsd and Ctss was regulated by Mmp9 and Fth1 (Fig. 5B).

Validation of the hub genes by qPCR and western blot

To validate the accuracy of the omics data and the hub genes in PPI networks, 12 adult mice were treated with laparotomy (Sham group, n = 6) and cecal ligation and puncture (T group, n = 6). Firstly, qPCR of three hub genes like Tpt1, Mmp9 and Fth1 were performed between SIC and healthy samples. As shown in Fig. 6A, the high expression of Tpt1, Mmp9 and Fth1 in SIC samples were detected, consistent with our transcriptome data. The high protein expression level of TPT1, MMP9 and FTH1 had been verified again in the western blot experiment, which was consistent with the qPCR results (Fig. 6B).

Figure 6 Verification of the expression level of TPT1, MMP9, and FTH1.

(A) qPCR results of the three genes. (B) Western blot of the three proteins. **p < 0.01.

TPT1 is involved in the development of SIC

Hl-1 myocardial cell line of mice was induced by LPS (25 mg·L−1) for 24 h to detect the protein expression. Western blot results (Fig. 7A) showed that the expression of TPT1 protein increased significantly in the process of LPS-induced SIC, indicating that TPT1 may be positively correlated with the development of SIC. Cell cycle flow cytometry showed that LPS significantly inhibited cells in G2/M phase, which was reversed by TPT1 silencing (Fig. 7B).

Figure 7 TPT1 is involved in the development of SIC.

(A) Western blot results of TPT1 and GAPDH in the HL-1 cell line under the treatment of LPS. (B) Cell cycle profile results of control, LPS-24H, LPS+NC, and si-TPT1 groups.

In order to detect the effect of TPT1 on cell apoptosis, TUNEL was performed based on the HL-1 cells induced by LPS. Results showed that the LPS group significantly increased apoptosis, while the si-TPT1 group reversed apoptosis (Fig. 8).

Figure 8 TUNEL results.

TUNEL results of the HL-1 cells induced by LPS in control, LPS-24H, LPS+NC, and si-TPT1 groups.

Discussion

SIC is a reversible cardiac dysfunction induced by sepsis. The mortality rate of sepsis combined with SIC is significantly increased (Vieillard-Baron, 2011). However, there is no consensus on the definition, diagnosis and treatment of SIC. In this study, we used the expression data about SIC mice to identify the mRNAs involved in transcriptome alteration. Finally, three hub genes and corresponding proteins were verified by qPCR and western blot.

Scholars gradually focus on the mechanism of sepsis cardiomyopathy. Romero-Bermejo et al. (2011) indicate that SIC may be induced by ischemia caused by insufficient coronary artery blood flow. However, Cunnion et al. (1986) discovered that the coronary blood flow velocity in sepsis cardiac dysfunction patients was not less than that in patients with normal cardiac function by monitoring the coronary blood flow in sepsis patients. These findings indicated the heterogeneity of coronary blood flow in SIC and away from a direct link between decreased coronary blood flow velocity and the development of the condition (Krishnagopalan et al., 2002). Unlike an obstructive coronary disease, sepsis cardiomyopathy is not accompanied by large myocardial infarction but manifests as reversible myocardial dysfunction (Thygesen et al., 2007). Many studies have confirmed that chemical mediators such as nitric oxide, endotoxin and inflammatory cytokines play a certain role in SIC (Kong et al., 2017). Current studies suggest that sepsis cardiomyopathy may be a pathological process involving various comprehensive factors, including cardiac microcirculation, myocardial inhibitory factors, mitochondrial dysfunction and calcium ion homeostasis imbalance (Manolis et al., 2021).

Controlled tumour protein (Tpt1) is translationally involved in cell proliferation, migration and apoptosis, as well as sugar and lipid metabolism. In recent years, researches based on Tpt1 gene mainly focused on cancer cells (Tuynder et al., 2002). In our study, Tpt1 was the hub gene according to the PPI results and TF-miRNA-mRNA network. In addition, our experiments confirmed that TPT1 protein expression is enhanced with the induction of SIC in mice. Silencing TPT1 protein significantly reversed apoptosis and LPS-induced cell inhibition in the G2/M period. All the results indicated that the expression of TPT1 was positively correlated with SIC. However, in previous studies, there was nearly no evidence showed that Tpt1 correlated with SIC, therefore, many clinical samples and high-throughput data need to be carried out in this field in future. Matrix metalloproteinases, also known as stromal proteins, are zinc-dependent endopeptidase enzymes that break down laminin, collagen and fibrinogen found in the extracellular matrix (Theocharis et al., 2016). MMPs family has at least 26 members, with MMP2 and MMP9 being the most closely associated with angiogenesis. MMPs can not only degrade ECM but also destroy the integrity of the blood-brain barrier in cerebral ischemia, other traumatic brain injuries and tumor (Asahi et al., 2001). The activity of MMPs increase in various cardiovascular diseases, including acute or chronic heart failure and atherosclerosis (Jiang, Jiang & Tao, 2013). As an important member of MMPs, MMP9 can effectively degrade Type IV collagen, the main component of the basement membrane and degrade the endothelial basement membrane, leading to the decrease of endothelial barrier function (Rowe & Weiss, 2008). In our results, the high expression level of transcript and protein of Mmp9 were detected in SIC group, which is concordant with a previous study in which the cardiac MMP-2 and MMP-9 were positively involved in cardiac heart rate and negatively correlated with the Left Ventricular Stroke Work Index (LVSWI). Moreover, increased activity correlates positively with myocardial apoptosis (Wohlschlaeger et al., 2005). The results of Zhang et al. (2018) also show that levels of α-SMA and MMP-9 elevate significantly in septic mice, indicating that MMP-9 mediates the differentiation of cardiac fibroblasts. Florence Morriello indicates that MMP-9 and HMGB-1 could be biomarkers for predicting the severity of cardiac dysfunction in septic patients (Florence Morriello et al., 2007). The above studies once again proved the potential of MMP-9 as a biomarker of SIC.

Ferritin heavy chain polypeptide 1 (FTH1) is a widely studied and mature reporter gene of magnetic resonance imaging (MRI), which has been widely used in the tracer study of various types of cells (Zhuo et al., 2019). Most studies have shown that FTH1 expression is not affected by cell division and proliferation because of its integration into the cell genome (Chan et al., 2018). In our results, FTH1 was involved in the PPI and TF-miRNA-mRNA networks and was verified by qPCR and western blot. Li et al. (2020) defined that the mRNA levels of FTH1, S100A9 and TYROBP were significantly increased in coronary atherosclerotic heart disease (CHD) patients. The negative effect of FTH1 downregulation on cardiac function is identified by a previous report showing that mice lacking FTH1 in cardiomyocytes obtain increased oxidative stress, resulting in mild cardiac dysfunction upon aging (Fang et al., 2020). A circRNA-miRNA-mRNA regulatory network that contained FTH1 was successfully constructed by identifying DEGs related to iron metabolism in the myocardial tissues of pressure overload-induced heart failure mice (Zheng et al., 2021). Previously, no study has directly linked FTH1 to SIC; but based on our result, there may be a connection between FTH1 and the onset and progression of cardiomyopathy.

Conclusion

In this study, we successfully identified one set of genes that are consistently up-regulated and one that is consistently down-regulated between the healthy and SIC groups. A total of 221 down-regulation and 342 up-regulation expressed genes were obtained in two clusters and were selected for further analysis. The selected genes Tpt1, Mmp9 and Fth1 were verified with qPCR and western blot, and the results were consistent with the expression level in transcriptome data. Western blot results showed that TPT1 protein is enhanced with induction of SIC in mice. Silencing TPT1 protein significantly reversed apoptosis and LPS-induced cell inhibition in the G2/M period. Our findings provide basic insights into the molecular mechanism of SIC.

Supplemental Information

Supplemental Information 1 ARRIVE_Checklist.

Click here for additional data file.

All the authors wish to acknowledge NCBI Gene Expression Omnibus (GEO) database for its assistance.

Abbreviations

SIC Sepsis-induced cardiomyopathy

GEO Gene Expression Omnibus

GO Gene Ontology Database

ORF open reading frame

CC Cellular component

MF Molecular function

BP Biological process

KEGG Kyoto Encyclopedia of Genes and Genomes

NCBI National Center for Biotechnology Information

Additional Information and Declarations

Competing Interests

Author Contributions

Ethics

Data Availability

The authors declare that they have no competing interests.

Dehua Liu analyzed the data, authored or reviewed drafts of the article, and approved the final draft.

Tao Wang performed the experiments, prepared figures and/or tables, and approved the final draft.

Qingguo Wang performed the experiments, prepared figures and/or tables, and approved the final draft.

Peikang Dong performed the experiments, prepared figures and/or tables, and approved the final draft.

Xiaohong Liu analyzed the data, authored or reviewed drafts of the article, and approved the final draft.

Qiang Li analyzed the data, authored or reviewed drafts of the article, and approved the final draft.

Youkui Shi analyzed the data, authored or reviewed drafts of the article, and approved the final draft.

Jingtian Li analyzed the data, prepared figures and/or tables, and approved the final draft.

Jin Zhou conceived and designed the experiments, authored or reviewed drafts of the article, and approved the final draft.

Quan Zhang conceived and designed the experiments, authored or reviewed drafts of the article, and approved the final draft.

The following information was supplied relating to ethical approvals (i.e., approving body and any reference numbers):

This research involved mice experiments, which the Ethical Committee of the Affiliated Hospital of Weifang Medical University approved.

The following information was supplied regarding data availability:

The raw data is available at figshare: Zhang, Quan (2023). SIC bioinformatics and experiments for Peer J. figshare. Dataset. https://doi.org/10.6084/m9.figshare.23994084.v1.

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
