# Peer review of "Identification of key genes in sepsis-induced cardiomyopathy based on integrated bioinformatical analysis and experiments in vitro and in vivo"

_PeerJ, doi:10.7717/peerj.16222_

## Round 0.1 · original submission · Major Revisions

a point-to-point response letter is needed

Reviewer 1 ·

Basic reporting

1.In the abstact section the method should describe comprehensively, the Results describe more comprehensively and more details
2.The introduction logic is poor and need modified . I also suggest that you add the incidence rate of sepsis-induced cardiomyopathy (SIC). The main text abbreviation and abstract abbreviation are separated, in the introduction section SIC abbreviation needs list full names.
3.The method needs more details
What kind of mouse is mice,mice weighing 250 to 300 g ? Please list PCR primer sequences and PCR cycling conditions. “The primary antibody 145 MAP2K1/MEK1 antibody (NBP2-67358, Novus Biologicals) and CCL25/TECK antibody (MAB3341, Novus 146 Biologicals) “inconsistent with Tpt1, Mmp9, and Fth1.
4.Please add the scale bar of TUNNEL .
5.In Figure 1,Cluster 1 and Cluster3 were analyzed,but at the bottom,the explanation was changed into Cluster 2 and Cluster3.
6.In Figure 6, The WB is not clear,please change one。In Figure 7,there is no grouping informationIn. In Figure 8,there was no scale bar.
In method Western blot, There was no information related to Sitpt1, including which company synthesized, the relevant sequences and verify the inh

Experimental design

.

Validity of the findings

.

Reviewer 2 ·

Basic reporting

Overall, the authors' experimental approach is clear and the logical flow is smooth. However, they did not delve deep into the specific mechanisms and downstream pathways of the identified target genes. Further investigation and discussion on this aspect would have been beneficial. Moreover, although the authors validated the role of TIP1 in septic cardiomyopathy through flow cytometry and TUNEL experiments, they did not validate the specific mechanisms of the other two genes, Mmp9 and Fth1.

Experimental design

1.The capitalization of "the" is incorrect in line 23.
2.The capitalization of "mRNA" is incorrect in the annotation for Figure 4, caption A.
3.Figure 7, panel A, does not provide the experimental conditions (corresponding groups are not indicated), and the legend for Figure 7, panel B, is too rough and not aesthetically pleasing.
4.Figure 8 does not provide the corresponding staining names, such as DAPI, TUNEL, etc.

Validity of the findings

In line 114, it should be noted that a smaller p-value indicates a higher probability of having a correlation, rather than a stronger relationship.
In this article, the authors aimed to uncover the genetic mechanisms and suitable biomarkers behind septic cardiomyopathy. They achieved this by mining existing public data, conducting gene temporal analysis, protein interaction network analysis, GO&KEGG analysis, and predicting miRNA-mRNA interactions. Additionally, they constructed an animal model to further validate their hypotheses and investigate the key genes involved in septic cardiomyopathy.

---

## Round 0.2 · accepted · Accept

The authors have addressed all questions.